# Spatiotemporal and Individual Patterns of Domestic Cat (*Felis catus*) Hunting Behaviour in France

**DOI:** 10.3390/ani13223507

**Published:** 2023-11-14

**Authors:** Irene Castañeda, Marie-Amélie Forin-Wiart, Benoît Pisanu, Nathalie de Bouillane de Lacoste

**Affiliations:** 1Ecology and Genetics of Conservation and Restoration, UMR INRAE 1202 BIOGECO, Université de Bordeaux, 33615 Pessac, France; irene.castaneda-gonzalez@u-bordeaux.fr; 2DataEthoEco, 3 Chemin de Touriac, 33480 Sainte-Hélène, France; ma.forinw@hotmail.fr; 3UAR Patrimoine Naturel (Office Français de la Biodiversité (OFB/MNHN)), 36 rue Geoffroy Saint-Hilaire, CP41, 75005 Paris, France; benoit.pisanu@ofb.gouv.fr; 4SFEPM (Société Française pour l’Etude et la Protection des Mammifères), 19 Allée René Ménard, 18000 Bourges, France

**Keywords:** domestic cat, citizen science, *Felis catus*, predation, prey brought home, seasonality, climatic factors, human footprint, individual variability

## Abstract

**Simple Summary:**

Domestic cats prey on many small animals throughout the year. However, the role of the environment and season on this hunting behaviour still needs to be clarified. Based on a citizen science project, we assessed the role of age and sex, the seasons, and the degree of human-related environmental degradation on the number of preys returned home on a monthly basis over 8 years by 5048 pet cats in France. Prey (n = 36,568) were mostly represented by small mammals (68%; voles, mice, and shrews), birds (21%; passerines), and reptiles (8%; lizards). More shrews, birds, and reptiles were brought home by young cats. Shrews peaked in summer, rodents in summer–autumn, birds in spring–summer and autumn, and lizards in spring and summer. The number of voles and mice increased where human degradation pressure was low, and conversely, lizards and birds increased where this pressure was high. Rainfall played a minor role, and cats caught animals according to their distributional geography (e.g., lizards in southern regions). Bearing in mind that the number of preys brought home underestimates the true number caught, we need now to evaluate the amount of prey available in a cat’s home range, and how many are really caught within, to fully understand predation impact.

**Abstract:**

Domestic cats (*Felis catus*), one of the most popular pets, are widespread worldwide. This medium-sized carnivore has well-known negative effects on biodiversity, but there is still a need to better understand the approximate causes of their predation. Based on a citizen science project, we assessed the role of spatiotemporal (i.e., latitude, longitude, and seasons), climatic (i.e., rainfall), anthropogenic (i.e., human footprint, HFI), and individual (i.e., sex and age) variables on the number of preys returned home by cats in metropolitan France. Over the 5048 cats monitored between 2015 and 2022, prey from 12 different classes (n = 36,568) were returned home: 68% mammals, 21% birds, and 8% squamates. Shrews brought home by cats peaked during summer, while rodents were recorded during summer–autumn. Birds brought home by cats peaked in spring–summer and in autumn, and lizards peaked in spring and in late summer. Lower HFI was associated with more voles and mice brought home, and the opposite trend was observed for lizards and birds. Younger cats were more prone to bring home shrews, birds, and reptiles. Although environmental factors play a minor role in prey brought home by cats, some geographical characteristics of prey species distribution partly explains the hunting behaviour of cats.

## 1. Introduction

Wild African cats (*Felis silvestris lybica*) were domesticated in the Middle East more than 9000 years ago [1]. Modern-day domestic cats (*Felis catus*) are opportunistic and generalist predators that are considered one of the 100 worst invasive species in the world, threatening many mammals, birds, reptiles, and amphibians [1,2,3]. Across the globe, the number of cats ranges between 600 million to 1 billion; for instance, in European countries, 91 million households currently possess approximately 127 million cats [4,5]. Crowley et al. [6] distinguish different cat populations ranging from feral cats that are neither dependent on nor controlled by humans to indoor cats that are fully confined to their food, reproduction, and movements being closely controlled by humans. In France, the domestic cat is the most popular pet with more than 15 million individuals registered in 2020 (i.e., fully indoor and indoor–outdoor cats, I-CAD database 2020), but this number can be doubled when considering free-ranging semi-owned individuals (e.g., farm cats) [7].

Cats’ effects on biodiversity are diverse including their predation, competition, behavioural disturbance, disease transmission, and hybridisation [8,9,10]. Cat predation is a well-documented phenomenon at both local [5,9,10] and continental scales [10,11,12,13]. Cats can locally reduce mainland vertebrate populations including birds and mammals [10,11] as well as invertebrates [14]. In countries where large-scale estimations exist for other direct mortality sources (e.g., collisions with windows, vehicles, or transmission lines), cats far exceed these other sources of human-related mortality (excluding indirect factors like habitat loss, e.g., industrial and other human activities) [15,16].

Unlike feral cats, house-based free-ranging cats (i.e., indoor–outdoor cats) are provided with medical care and shelter by owners, so they are not subjected to diseases or fluctuations in resources and are therefore able to exceed environmental carrying capacity [17,18]. If unowned cats, as opposed to owned pets, cause the majority of the mortalities induced by cat predation [9], indoor–outdoor cats still cause substantial wildlife mortality, and studying their hunting behaviour is useful to improve our knowledge about the role of the domestic cat population on ecosystems. They frequently kill wild animals without consuming them and bring them to their owners [5,17,19]. While monitoring the number and diversity of preys brought home can provide a global account of the variation in the species caught by cats at a large spatial scale [10,18,20,21,22,23,24]; the fate of prey (i.e., directly eaten, left uneaten, or brought back) depends on the nature of the prey itself [25], and less than a third of prey caught are brought home [25,26].

Citizen science is a useful tool for understanding ecological issues [27] and is often used to assess the effects of domestic cat predation on wildlife communities [5,21,22,23,24,28]. In this study, we used data on prey brought home by free-ranging domestic (i.e indoor–outdoor) cats in France recorded by the citizen science project named “Chat domestique et Biodiversité” (En: domestic cats and biodiversity) led by the French society for the study and protection of mammals (SFEPM) and the National Museum of Natural History of Paris (MNHN). First, we quantified the number of preys brought home by cats. Second, we attempted to analyse changes in prey brought home by cats in relation to temporal (seasons), climatic (rainfall), biogeographic (latitude and longitude), anthropogenic (Human Footprint Index, HFI), and individual variables (age and sex). We used the results to test the following predictions:Following Thomas et al. [28] in the UK, we predict that the number of prey brought home by domestic cats will be higher during spring and summer coinciding with prey breeding seasons in temperate areas than during the rest of the year.Weather conditions strongly influence small mammals [29,30,31], birds [32,33], and lacertid [34] activity in Europe; rainfall positively influences small mammal activity but negatively influences the activity of birds and lacertids. Thus, we predict a higher number of small mammals brought home by cats in regions with a higher relative rainfall, while the number of birds and lacertids brought home by cats in such localities will be lower.Biogeographic factors determine prey species’ ecological range [35,36]. Accordingly, we predict that the number of individuals of endothermic prey (i.e., mammals and birds) brought home by cats will increase from the southeast to the northwest while the opposite pattern will be true for ectothermic prey (i.e., lacertids) brought home by cats.In line with other European studies [37,38], we predict that locations with higher HFIs—an index of human activity affecting habitat—will be related with a low number of preys brought home by cats.Individual characteristics of cats such as sex and age have been linked with predation rate [18] and the type of prey captured [35]. Accordingly, as found by Kauhala et al. [39], we predict that the diversity of prey brought home by young cats will be higher than that for adult ones.

## 2. Materials and Methods

### 2.1. Preys Brought Home

The project ran from 2015 to 2023. Volunteering cat owners joined the citizen science survey to record preys brought home by their cats through the website: https://www.chat-biodiversite.fr/. People were firstly recruited through a mailing list specifically addressed to members of the French Society For The Study and Protection of Mammals (SFEPM) and to a network of related associations. The project was supported by the French National Museum of Natural History and the Bird Protection League. It also benefited from communications by the French Society for Herpetology and the SFEPM to gain in visibility. When they signed up to the project, cat owners informed if they were able to survey the predation of their animal (either regularly or occasionally). Cat owners also reported if they were observers uninitiated to the identification of species (no naturalist skills), those with an intermediate level (recognition of main species groups), or confirmed naturalists (species recognition skills). This information was used for uncertain species determination, especially when no picture of the prey was available. Prey identification was only used at the order level. On the website, volunteers recovered opportunistic or systematic predation events of their cats. Also, volunteers provided information about their cat (e.g., name, sex, birth date, and breed) and about prey brought home (e.g., the date of the capture, the localization, and the species or at least the species group). To help volunteers to identify prey, a photographic guide of potential cat prey is available on the website. Most of the monitored cats in this study were described by their owners as non-pedigree individuals (93%, i.e., generic domestic shorthair or longhair cats); thus, we did not include this variable in our analysis.

### 2.2. Spatial Variables

We created a 100 m circular buffer around each owner location (as domestic cats generally remain close to their owner’s home [40]) to estimate the mean annual temperature, mean annual rainfall, and HFI (Human Footprint Index). Mean annual temperature and rainfall at 5 m resolution were sourced from the WorldClim2 dataset (www.wordlclim.org, accessed on 6 February 2023 [41]). We quantified anthropogenic influence using the HFI layer version 2, 1995–2004 (Wildlife Conservation Society—WCS 2005); this database is a global spatial dataset of the HFI normalized by biome and realm. Global HFI is calculated using population density, human land use, infrastructure (e.g., night-time lights, built-up areas), and human access (e.g., railroads, roads, and coastlines). This index, rated on a scale of 0 (minimum) to 100 (maximum) for each terrestrial biome, is a quantitative analysis of human influence across the globe. A score of 1 indicates the least human influence in the given biome.

### 2.3. Statistical Analysis

The analyses were performed on the sum of all prey items reported for each cat in a given month between January and December across all years. A total of 40,456 preys brought home were recorded between January 2015 and August 2022, of which 39,085 belonged to an identified owner. Overall, from these observations, 2.8% (n = 1097) of prey records were not localized, yielding in 37,984 spatially and temporally referenced prey records, of which 17 were discarded because they corresponded to locations outside the metropolitan French territory or were aberrant due to erroneous input by owners. A total of 244 of these records were not available because of an absence of precise description of the preys returned home. Within the dataset comprising 37,723 full records of prey brought home by cats, 11 records from unclearly identified cats were discarded. The dataset then corresponded to 5048 unique identified cats from 4095 owners, for which 37,711 records of prey return were noted across the entire French metropolitan area. For modelling purposes, 495 records without HFI information, 320 records without cat age, and 328 records belonging to senior individuals (minorities in the dataset) were discarded, reducing the dataset to 36,568 records (Figure 1). The anonymized dataset is available in Appendix A.

Generalized additive mixed models (GAMMs, [40]) with a Poisson error distribution and a log link were used to analyse the variability in the five main prey categories brought home by cats (i.e., soricids, cricetids, murids, passeriformes, and lacertids). These models allow us to characterize nonlinear relationships and to detect minimum, maximum, and inflexion points and threshold values. These variations are expressed by the number of effective degrees of freedom (“edf”) estimated by the models. An edf value of 1 is equivalent to a linear relationship; an edf greater than 1 and less than 2 indicates a weak nonlinear relationship; and when the edf is greater than 2, the relationship is strongly nonlinear [42].

We used seven explanatory variables in each GAMM: month (discrete variable, range: 1–12, scaled centred), HFI (continuous variable, scaled centred), mean annual rainfall (continuous variable), latitude and longitude coordinates (continuous variable, transformed into metric coordinates in the Lambert II extended projection system), age (continuous variable, range: 0–14), and sex (categorial variable with two levels: female, male). The effect of month was fitted with a cyclic cubic regression spline following Krauze-Gryz et al. [21], while the effects of HFI, rainfall, and age by sex were fitted with a cubic regression spline. The effect of latitude and longitude was investigated by producing a full tensor product smooth especially useful for representing functions of covariates.

In each GAMM, cat identity, county, and year of observation were included as random effects and fitted with a ridge penalty spline. Indeed, the significant variability in the data collection effort between 2015 and 2022 is related to the history of the project, with years with little or no activity resulting in less information on prey brought home. Similarly, each cat has its own profile of hunting behaviour, with cats for which many predation records have been made sometimes being particularly focused on a group of small mammals, whereas most monitored cats have only one predation record in the whole project. Additionally, we assumed that cats living closely in space (i.e., at the county scale), would have more similar prey species brought home than cats living at larger distances from each other.

The collinearity between explanatory variables was investigated, so the mean annual rainfall variable was preferred to the temperature variable. Full models were validated by graphic inspection following Zuur et al. [43]. All analyses were performed in the R 4.2.2 environment (R Core Team 2022) with RStudio 2022.12.0 (RStudio Team 2022), using ‘mgcv 1.8-41’ (bam function, [44]) and ‘mgcViz 0.1.6’ [45] packages and codes provided by Zuur et al. [42].

## 3. Results

### 3.1. Prey Species Brought Home by Cats

Over the 8-year survey period, 3073 cats’ owners only reported prey returned home once, while 2610 cats’ owners reported at least two preys, of which 812 were very active, reporting at least 10 prey items (Figure 2). The mean (±SD) length of time owners reported over was 3.1 months ± 5.0 months. The mean (±SD) number of individual cats followed by season was 2048.5 individuals ± 551.1. The mean (±SD) number of preys brought home by individual cats over the seasons was 4.5 preys ± 0.5.

During the survey, cats brought home prey belonging to 12 different classes (Actinopterygii, Amphibia, Annelida, Arachnida, Aves, Chilopoda, Clitellata, Gastropoda, Hexapoda, Malacostraca, Mammalia, and Reptilia). The main prey species caught by cats were mammals (68.3%), followed by birds (21.4%) and squamates (8.4%). Among mammals, rodents represented 78.8% of the prey, followed by eulipotyphla (i.e., moles, shrews, and hedgehogs) with 15.8%. Among the eulipotyphla, the vast majority were shrews and shrew-like creatures; the most commonly brought-home species was the Eurasian Common Shrew (*Sorex araneus*) (accounting for 1% of the total mammals, while over 12% were ‘unidentified shrews’). The remaining percentages are attributed to lagomorphs (2.4%), chiropterans (1.3%), and carnivores (0.3%), while 1.3% of prey were identified as mammals.

If we consider only “small mammals” (Cricetidae, Gliridae, and Muridae), shrews, and related species (Soricidae) (i.e., 23,850 prey items), the most frequently reported species was the House Mouse (*Mus musculus*: 14.5%), followed by the Common Vole (*Microtus arvalis*: 3.7%) and the Wood Mouse (*Apodemus sylvaticus*: 3.5%). However, 64.9% of the small mammal data belong to categories not identified at species level, such as “field mice”, “shrews”, “voles”, and “small rodents (mice, field mice, voles)”, which account for 16.7%, 12.8%, 14%, and 21.4%, respectively, which clearly illustrates the difficulty of identifying these species.

Among birds, 83.3% of the species caught by the cats were Passeriformes, mainly the House Sparrow (*Passer domesticus*: 14.4%), European Robin (*Erithacus rubecula*: 9.3%), Eurasian Blackbird (*Turdus merula*: 8.3%), Eurasian Blue Tit (*Cyanistes caeruleus*: 6.8%), and Chickadee (*Parus major*: 6.3%), while 10.2% were unidentified passerines. The remaining percentages were mainly attributed to Columbiformes (5.8%) (including the Eurasian Collared Dove *Streptopelia decaocto*: 3.1% and Common Wood Pigeon *Columba palumbus*: 1.2%), Piciformes (0.5%), Caprimulgiformes (0.3%), Galliformes (0.3%), and Gruiformes (0.2%), while 9.3% were unidentified birds.

Among the squamates (<0.1% were unidentified), 92.5% of the species reported by the cats belonged to the Lacertilia family (18.3% were unidentified), including the Common Wall Lizard (*Podarcis muralis*: 56.7%), the Slow Worm (*Anguis fragilis*: 6.3%), and the Common Wall Gecko (*Tarentola mauritanica*: 5.2%). The remaining percentages are attributed to Serpentes (7.3%) and Scincomorpha (0.1%).

Only the following five main categories of preys (identified at least at the order level) were used in analysis: murids (Muridae-identified or not, i.e., 14,196 prey items), cricetids (Cricetidae-identified or not, i.e., 5149 prey items), soricids (Soricidae-identified or not, i.e., 3482 prey items), passerines (6486 prey items), and lizards (2882 prey items).

### 3.2. Seasonality of Prey Brought Home by Cats

Season was a significant predictor of prey brought home by cats (Table 1 and Figure 3). For each group, the relationship was highly nonlinear. Soricids were more commonly brought home by cats from spring to autumn, and their numbers increased until they reached a peak in summer. Cricetids were brought home throughout the year, but the number of reported preys reached a plateau in the summer. Murids were brought home by cats from spring to autumn, and their numbers increased until reaching a maximum in autumn. Birds were most commonly brought home by cats in late spring and early autumn. Lacertids were more frequently brought home by cats in spring and summer.

### 3.3. Climatic and Geographic Effects on Prey Brought Home by Cats

The mean annual rainfall was a significant predictor of the number of soricids (linear relationship) and cricetids brought home by cats (Table 1 and Figure 4). Their number increased with increasing rainfall, while the splines for murids, birds, and lacertids were not significant (Table 1).

The geographic distribution was a significant predictor of the types of preys brought home by cats (Table 1 and Figure 5). Soricids were more often brought home by cats in the northwest of France, up to twice as often as in the southeast. Cricetids were more often brought home in the northern part of France with a clear trend for the northeast, more than twice as many as in the southeast. Murids were preferentially brought home by cats in the north-western part of the country.

Cats tended to catch passerines fairly uniformly in metropolitan France (except in Corsica) and more particularly in south-western and central-eastern France. Lacertids were mostly reported in the south of France, including Corsica, up to three times more than in the north.

### 3.4. Anthropogenic Impacts on Prey Brought Home by Cats

The Human Footprint Index (HFI) had no influence on the number of soricids caught by cats (Table 1 and Figure 6). Fewer cricetids and murids were brought back home when the HFI was high, while the opposite pattern was observed for passerines and lacertids.

### 3.5. Individual Factors Influencing Prey Brought Home by Cats

The age and sex of individual cats were significant predictors of the types of prey brought home by cats (Table 1 and Figure 7). Soricids were predominantly reported by owners of young individuals (<2 years old), with the number decreasing rapidly with age. There was a significant difference in the number of cricetids brought home by females with a first peak for young adults (i.e., 2 to 4 years old) and a second, larger peak for adult females (i.e., 8 to 12 years old). The spline for males, whatever their age, was not significant. Females between the ages of 3 and 7 years tended to bring back home more murids, while the number of murids increased progressively until the age of 10 years for males. Passerines were mostly caught by young individuals, their numbers decreasing more gradually for females. A similar pattern was observed for lacertids, with the number of lacertids decreasing linearly for males with age.

## 4. Discussion

The temporal and spatial coverage of the present study—based on a citizen science program—far outweighs any previous study investigating the hunting behaviour of domestic cats. More precisely, cat owners regularly monitored the prey brought home by their cats for eight years, which allowed us to study the seasonal patterns of cat predation at the national scale. Moreover, the spatial coverage of our study includes almost all habitats representing metropolitan France, allowing us to assess the effects of prey brought home by cats nationwide.

Our results confirm that the prey brought home by cats were mainly small mammal species, followed by birds and reptiles, with a seasonal pattern of cat hunting behaviour depending on the nature of the prey itself. The most important hunting behaviour was from mid-spring to mid-summer, but also from late summer to mid-autumn. Such behaviour was only partially related to environmental factors for specific taxonomic groups of small mammals (shrews and voles). The age and sex of cats also influenced their hunting behaviour depending on the nature of the prey itself, with younger cats bringing home shrews, passerines, and lizards more often.

### 4.1. Prey Species Brought Home by Cats

The proportion of vertebrate prey brought home by cats in France is in accordance with that in previous studies at national or regional scales in Europe (Poland, UK: [17,23,37], Finland: [39], and Italy: [5]), Australia [46,47], and China [12], showing that small mammals were the principal prey brought home followed by birds and lizards. Similarities among the prey brought home by cats in European, Asian, and Australian studies may be related to preys’ relative availability in the studied areas, but also to a common evolutionary history of cats from those continents [48].

Our results, however, contrast with findings in New Zealand where rodents and insects [20], macroinvertebrates and rodents [22], or birds and rodents [49] were the main prey categories brought home by cats. The relative discrepancies between the rates of preys brought home among these former studies and the present one most probably lies in the difference in the number of monitored cats used to calculate the rates of preys brought home. For instance, Van Heezik et al. [49] monitored 151 cats, Morgan et al. [20] and Gillies and Clout [22] monitored around 80 cats each, while in the present study, we monitored 5048 cats. Comparing the results of such studies can also be challenging, given the range of predictors used and the different taxonomic levels involved (i.e., major taxonomic groups, families, species). The use of combinatorial probabilities as described by Murphy et al. [11] for diet analysis using stomach contents may facilitate comparisons between studies relying on prey species brought home.

Our study shows a very low number of bats reported (1.3%), while Ancilloto et al. [50] found that 28.7% of adult bats admitted to rehabilitation centres were attacked by cats in Italy. This underlines that predation by cats is still underestimated for bats and that more studies focusing on this group are necessary to clarify the impact of this predation [51]. It also raises the question of prey killed and reported and therefore accounted for, injured prey that does not survive and is not accounted for, and the fate of prey depending on its type [25].

### 4.2. Seasonality of Prey Brought Home by Cats

Cats exhibit a seasonal pattern of activity with greater home range sizes [52,53,54] and travel distances [7,55] during favourable seasons (i.e., spring and summer) in temperate areas. According to this seasonal cat behaviour, we found support for our hypothesis of a higher number of preys brought home by cats during spring and summer. This is in accordance with the findings of Krause-Gryze et al. [21], who found a seasonal increase in the cat predation of birds in rural habitats in Poland.

We found seasonal differences among the prey categories brought home by cats. For soricids, the seasonal pattern observed agrees with the one described by Krause-Gryze et al. [21] in rural habitats in Poland. These results are likely to reflect the seasonal reproduction and demography documented for *Sorex* spp. and *Crocidura russula* species in western European habitats where soricid populations experience an increase in reproductive activity from spring to summer and a decrease from late summer to late autumn (*Sorex* spp., Britain: [56,57]; Finland: [58]; *Crocidura russula*: [59]). In consequence, the soricids brought home by cats in France are mainly adults in the middle of reproduction period, likely affecting their population dynamics. Contrary to shrews, voles occurred from early summer throughout early autumn then decreased from mid-autumn until the next early spring, while for mice, we detected a continuous increase from mid-spring until early autumn and a decrease during winter. In contrast, in rural habitats in Poland, only an autumnal peak was described for rodents [21]. The rodents brought home by cats in France corresponds to the end of the breeding period observed during autumn and early winter for voles and mice in this country [60,61]. In consequence, cats not only predate rodents at the end of the breeding season when juveniles and subadults make up the bulk of the population at their peak density, but also affect reproductive adults during the breeding period in mid-spring.

Birds brought home by cats in France exhibited two peaks, one in late spring/early-summer and another one—less important—in mid-autumn. This spring peak in birds brought home by cats has already been observed in many habitats ranging from rural to urban or suburban [18,21,23,49]. Likely, these birds may be breeding adult males singing to attract a mate’s attention and/or defend their territory [62], while birds brought home during mid-autumn may be mainly juveniles. Only Kauhala et al. [38] found similar results to our autumn peak in Finland, where the highest number of birds brought home by cats occurred during autumn. In this case, cat predation’s impact on bird populations may be more important in autumn than in spring because the reproductive value of those surviving young individuals is higher at later periods of the year. Moreover, this result highlights that many common bird species populations may be at risk during autumn and that to detect this autumnal peak of birds brought home by cats, an important number of cats (>5000 individuals) over many years (>4 years) across a large geographical area (>540,000 km^2^) are needed.

Lacertids formed the third group of prey brought home by cats in our study, showing a bimodal pattern in the seasonal dynamics, composed of a first peak during spring (i.e., April–May) and a second peak during summer (i.e., August). The spring peak is in accordance with other studies from northern European countries such as in Finland [39] and in Poland [21]. Moreover, we detected a secondary peak during summer where lacertids may be still active and thus exposed to cat predation. Indeed, contrary to colder environments in northern European countries where lizards become inactive during this period, in France, *P. muralis* is active during early autumn (i.e., August–September) [63]. Also, *A. fragilis* populations located in the northwest Iberian peninsula are active for more than three quarters of the year [64]. Reptiles brought home by cats in France during the spring peak likely corresponds to adults emerging from hibernation and moving more frequently for reproduction (i.e., [65]), while the summer peak may represent juveniles, especially just after parturition [66].

### 4.3. Climatic and Geographic Effects on Prey Brought Home by Cats

Our results highlight the importance of considering the effects of rainfall on the number of preys brought home by cats across large spatial scales. Although rain reduces the range of activity of cats [7,54], the increase in small mammal activity [29,30,31,67] increases the probability of predator–prey contact. We found that the number of soricids and cricetids brought home by cats increased with increasing rainfall. This result suggests that under climate change scenarios, prey brought home by cats could shift with changing prey availability, particularly given the behavioural adaptability of cats. A number of predated prey shifting due to climate change has been demonstrated for other carnivores such as the Canadian lynx (*Lynx canadensis*) [68] or the Arctic fox (*Alopex logopus*) [69,70]. Studies regarding the dietary responses of cats support the generalist predation hypothesis by documenting a switch to alternative prey and an increase in diet diversity when their main prey abundance decreases [71,72,73]. Knowledge of the influence of climate variables on the availability of prey, and, hence, the number of predated preys, will improve our understanding of the synergetic effects of climate change and cat predation on wildlife.

Like the climate, biogeography also has strong effects on ecological community compositions [74], which may in turn influence what cats bring home [47,75]. In support of our predictions, endothermic prey (except birds) brought home by cats increased from the southeast to the north, while the opposite pattern was found for lacertids. These results may be due to the cooler temperatures in the north of France. Indeed, soricids’ distribution is favoured by cold weather conditions [76], while on the contrary, lacertids’ distribution is concentrated in warmer areas located in the Mediterranean Basin [77]. Cricetids brought home by cats increased from the southeast and southwest to the northeast, which may be due to the spatial distribution of their preferred habitat type (i.e., grasslands) in France [74,78,79]. Murids brought home by cats increased from the southeast to northwest. Because murids are composed of woodland species, this trend may be explained by the local habitat quality of woodlands embedded in the agricultural matrix and of nearby forests, as it has been shown for *A. sylvaticus* in the United Kingdom [80].

### 4.4. Anthropogenic Pressure Influence on Prey Returned Home

Our results partially support our prediction of a negative relationship between the number of preys brought home by cats and the HFI. We found that the number of small mammals decreased with increasing HFI, while the number of lacertids and birds brought home by cats increased. The decrease in the number of small mammals in highly anthropogenic areas is likely due to the decrease in diversity [81,82,83,84,85,86,87], abundance, and species richness [85,86].

Lacertids as ectotherm organisms are directly linked with the ambient temperature (e.g., [88,89,90]). Thus, it may be that lacertids are more likely to be brought home by cats in localities with higher HFIs due to the heat island effect [91,92]. Indeed, cat predation upon lacertids along anthropogenic gradients remains poorly studied in Europe [93,94]. Thus, knowledge about cat predation effects on this understudied prey category is essential in order to have a full overview of cat impacts on biodiversity.

The number of birds brought home by cats increased with increasing HFI. This result is in agreement with previous studies carried out in Northern Europe and North America, showing that the number of birds brought home by cats increases with increasing anthropogenic impacts [21,39,95]. The predation of these prey categories may be related to the availability of both birdfeeders and habitat suitability due to garden management.

### 4.5. Individual Factors Influencing Prey Brought Home by Cats

Hunting specializations in cats have been already described in Australia [24,96] and in France [7], as well as profiles of cat personalities responsible for wildlife predation [7,97,98,99]. Citizen science databases of prey brought home by cats collected at nation-wide scales over many years may provide the way forward to identifying persistent individual hunting patterns independent of the availability of prey in the environment.

In support of our hypothesis, greater numbers of soricids, lacertids, and passerines were brought home by young cats (i.e., <five years old) than by older ones. This result supports previous studies in which shrews, reptiles, and amphibians were more likely to be brought home by young cats, while older cats returned rodents [39]. Shrews produce vocalizations that are highly attractive to cats, but their decline with age shows that cats learn that they are inedible prey [39,100]. Indeed, species in the order Soricomorpha produce toxins in the saliva [101] to kill their prey [102], which may render them unpalatable to cats [103,104]. Young cats more often bring back lizards, probably because the movement of these species can stimulate their propensity to play. Predation on birds requires cats to be in a good physical condition (e.g., climbing on trees), which explains why these prey items are more likely to be brought by young cats [34]. Young and old females prefer cricetids, while older males choose mice. Further research into individual cat activity patterns would be helpful in order to confirm the patterns of the current study.

## 5. Conclusions

Our study was based on a large database benefiting from a large array of pet cats from many owners reporting the prey species seasonally brought home across a large area and over a long time. Age was a key parameter of the number and nature of prey brought home by cats, with younger individuals being more prone to bringing home shrews, birds, and reptiles. This result confirms that age is a key factor in cat predatory behaviour, which supports the hypothesis that reducing prey exposure to kittens may lessen their predatory behaviour as adults [105]. Our results also confirm seasonal patterns in the hunting behaviour of cats depending on the taxonomic nature of prey, which support earlier results that cats may prey upon many small vertebrate populations coinciding with their breeding periods. More importantly, the effect of human-related activity on the degradation of habitats has had opposite trends depending on the taxonomic nature of their prey, making small birds and lizards at greater risks to be prey in habitats highly degraded by humans such as urban areas. Although an abiotic factor, rainfall, played a minor role in explaining the prey brought home by cats, some geographical characteristics in prey species distribution partly explained the hunting behaviour of cats. Collectively, our study shows that preys are seasonally caught mostly by young cats (i.e., less than 5 years old) according to their species availability and depending on the quality of the environment within the home range of domestic cats. Still, bearing in mind that the number and species of preys brought home only reflects a part of the absolute quantity caught, we need to better evaluate both prey number availability and numbers caught in the home range of pet cats to better infer its impact. Such a goal could be achieved by linking both realistic cat predation rates from video loggers [25,26] and prey population metrics from modern methods of remote censuses based on acoustic or camera trapping [106].

## Figures and Tables

**Figure 1 animals-13-03507-f001:**
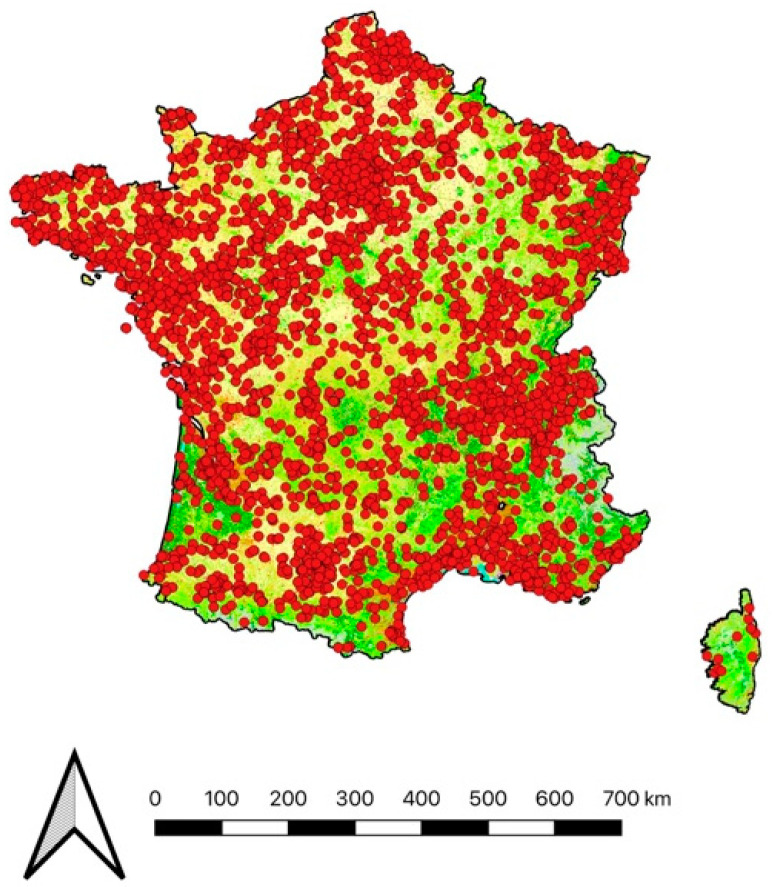
Location of the 36,568 preys brought home by cats over the 8-year period of survey. Reclassified Corine Land Cover 2018 in four categories: urban (grey), natural (green), agricultural (yellow), and water (blue).

**Figure 2 animals-13-03507-f002:**
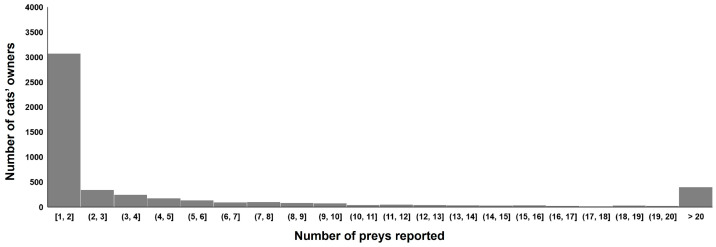
Distribution of the number of cat owners who recorded at least one prey item brought home by their cat during the study.

**Figure 3 animals-13-03507-f003:**
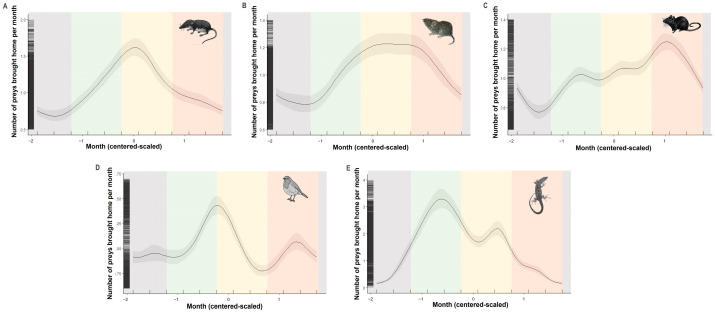
The predicted numbers of five main prey categories brought home per month by cats throughout the year: soricids (**A**), cricetids (**B**), murids (**C**), passerines (**D**), and lacertids (**E**). Curves represent cyclic splines fits with 95% confidence intervals as predicted by the GAMMs summarized in Table 1. Seasons are represented by transparent areas: winter (grey), spring (green), summer (yellow), and autumn (orange).

**Figure 4 animals-13-03507-f004:**
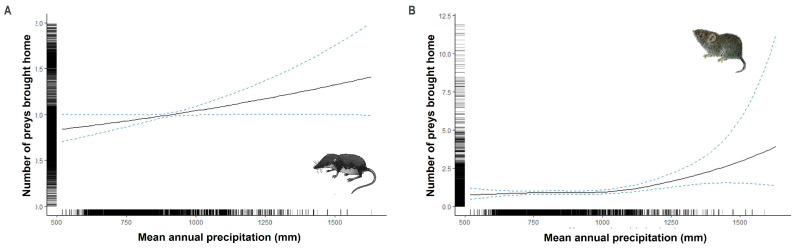
The predicted number of soricids (**A**) and cricetids (**B**) brought home by cats according to the mean annual rainfall. Curves represent cubic regression splines fits with 95% CI as predicted by GAMMs summarized in Table 1.

**Figure 5 animals-13-03507-f005:**
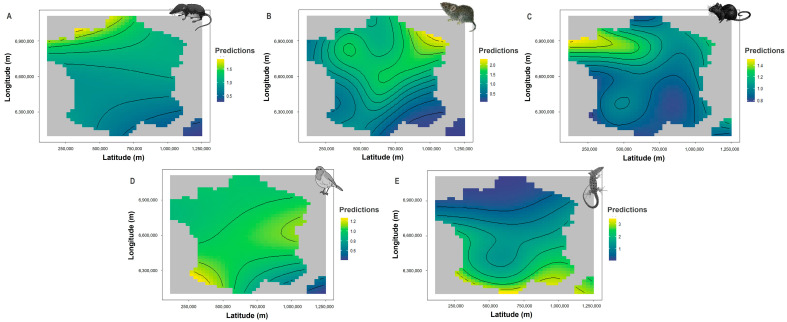
The predicted numbers of the five main prey categories brought home by cats according to latitude and longitude: soricids (**A**), cricetids (**B**), murids (**C**), passerines (**D**), and lacertids (**E**). A simple heatmap represents both variables and their interaction. The interior is a topographic map of predicted values with yellow to blue colours representing larger to smaller predictions.

**Figure 6 animals-13-03507-f006:**
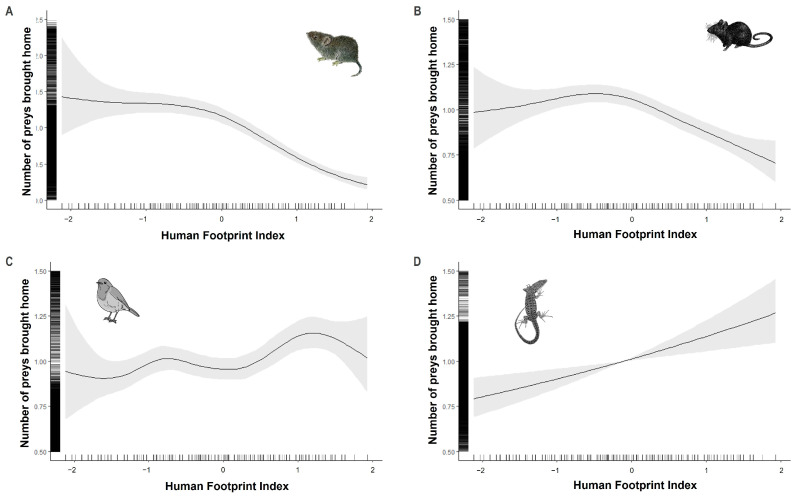
The predicted number of cricetids (**A**), murids (**B**), Passeriformes (**C**), and lacertids (**D**) brought home by cats according to the Human Footprint Index. Curves represent cubic regression splines fits with 95% CI as predicted by GAMMs summarized in Table 1.

**Figure 7 animals-13-03507-f007:**
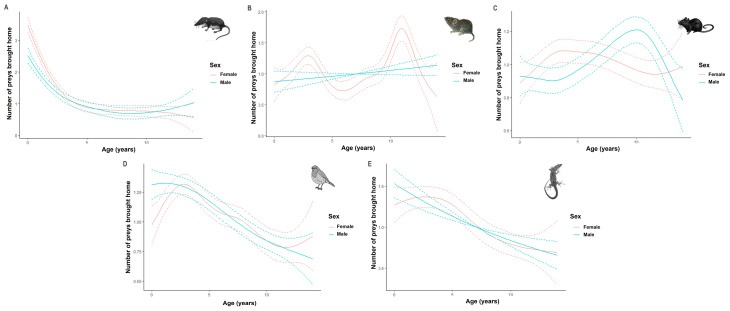
The predicted numbers of the five main prey categories brought home by cats according to their age and sex: soricids (**A**), cricetids (**B**), murids (**C**), passerines (**D**), and lacertids (**E**). Curves represent cubic regression splines fits with 95% CI (dashed lines) as predicted by GAMMs summarized in Table 1.

**Table 1 animals-13-03507-t001:** Summary of generalized additive mixed models (GAMMs) explaining the numbers of the five main prey categories brought home by free-ranging domestic cats. Estimated degrees of freedom (Edf) are given for splines.

Model	Predictor	Estimate (SE)	Statistic	*p*-Value *
Soricids R^2^_adj_ = 0.48 Pearson estimate = 0.66 Deviance explained = 53.6%	Intercept	−2.78 (0.24)	*z* = −11.52	**0.0001**
	Month	Edf = 5.33	X^2^ = 492.31	**0.01**
	Rainfall	Edf = 1.05	X^2^ = 3.97	**0.05**
	Latitude × Longitude	Edf = 5.37	X^2^ = 45.65	**0.0001**
	HFI	Edf = 3.12	X^2^ = 4.94	0.14
	Age _(Female)_	Edf = 4.21	X^2^ = 172.49	**0.0001**
	Age _(Male)_	Edf = 3.45	X^2^ = 86.96	**0.0001**
	CatID	Edf = 1154.40	X^2^ = 8888.47	**0.001**
	Town	Edf = 0.28	X^2^ = 2.53	0.74
	Year	Edf = 6.67	X^2^ = 827.85	**0.01**
Cricetids R^2^_adj_ = 0.41 Pearson estimate = 0.76 Deviance explained = 57.1%	Intercept	−2.16 (0.13)	*z* = −17.19	**0.0001**
	Month	Edf = 4.56	X^2^ = 571.14	**0.05**
	Rainfall	Edf = 3.12	X^2^ = 13.15	**0.01**
	Latitude × Longitude	Edf = 12.41	X^2^ = 82.82	**0.0001**
	HFI	Edf = 3.40	X^2^ = 163.78	**0.0001**
	Age _(Female)_	Edf = 7.50	X^2^ = 72.13	**0.0001**
	Age _(Male)_	Edf = 1.00	X^2^ = 2.38	0.12
	CatID	Edf = 1309.41	X^2^ = 16,494.29	**0.01**
	Town	Edf = 0.60	X^2^ = 90.78	0.85
	Year	Edf = 5.79	X^2^ = 544.12	0.82
Murids R^2^_adj_ = 0.37 Pearson estimate = 1.16 Deviance explained = 47.2%	Intercept	−0.65 (0.05)	*z* = −14.14	**0.0001**
	Month	Edf = 6.63	X^2^ = 209.70	**0.0001**
	Rainfall	Edf = 1.83	X^2^ = 1.63	0.52
	Latitude × Longitude	Edf = 1.30	X^2^ = 49.32	**0.0001**
	HFI	Edf = 2.84	X^2^ = 39.99	**0.0001**
	Age _(Female)_	Edf = 3.72	X^2^ = 10.44	*0.1*
	Age _(Male)_	Edf = 4.73	X^2^ = 30.14	**0.0001**
	CatID	Edf = 1828	X^2^ = 51,789.30	0.79
	Town	Edf = 0.00	X^2^ = 0.00	0.99
	Year	Edf = 4.97	X^2^ = 41.94	**0.05**
Passeriformes R^2^_adj_ = 0.18 Pearson estimate = 0.88 Deviance explained = 24.7%	Intercept	−1.16 (0.04)	*z* = −31.89	**0.0001**
	Month	Edf = 6.91	X^2^ = 251.62	**0.0001**
	Rainfall	Edf = 1.00	X^2^ = 1.69	0.19
	Latitude × Longitude	Edf = 5.62	X^2^ = 11.61	*0.1*
	HFI	Edf = 4.52	X^2^ = 18.14	**0.01**
	Age _(Female)_	Edf = 4.90	X^2^ = 46.01	**0.0001**
	Age _(Male)_	Edf = 2.71	X^2^ = 56.38	**0.0001**
	CatID	Edf = 1045.30	X^2^ = 6135.86	0.96
	Town	Edf = 0.20	X^2^ = 0.57	0.77
	Year	Edf = 0.80	X^2^ = 1.10	0.91
Lacertilians R^2^_adj_ = 0.46 Pearson estimate = 0.58 Deviance explained = 55.7%	Intercept	−3.06 (0.08)	*z* = −37.57	**0.0001**
	Month	Edf = 7.61	X^2^ = 785.08	**0.0001**
	Rainfall	Edf = 2.79	X^2^ = 3.59	0.31
	Latitude × Longitude	Edf = 1.86	X^2^ = 297.10	**0.0001**
	HFI	Edf = 1.00	X^2^ = 11.18	**0.0001**
	Age _(Female)_	Edf = 3.14	X^2^ = 29.88	**0.0001**
	Age _(Male)_	Edf = 1.00	X^2^ = 23.71	**0.0001**
	CatID	Edf = 949.90	X^2^ = 4953.17	**0.001**
	Town	Edf = 0.00	X^2^ = 0.00	0.95
	Year	Edf = 3.56	X^2^ = 8.90	*0.06*

* Significant effects are in bold, trends are in italic.

## Data Availability

The full dataset can be obtained on reasoned request from the corresponding author.

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
