# Peer review of "Spatiotemporal and Individual Patterns of Domestic Cat (Felis catus) Hunting Behaviour in France"

_animals, 2023, doi:10.3390/ani13223507_

Round 1
Reviewer 1 Report
Comments and Suggestions for Authors
Review of: spatio-temporal and individual patterns of domestic cat hunting behaviour in France.
This study reports on prey brought back home as reported by over 5,000 cat owners throughout France. It also relates relative numbers of the different taxa to a range of potentially explanatory variable. The results are interesting in that they provide a nation-wide assessment of relative catch rates of different taxa, however, in terms of variables describing variations in numbers of prey reported, there is no new information that hasn’t been published elsewhere before; it is just new for France. I find the logic of some of the predictions hard to follow.
The paper doesn’t address anywhere the issue of whether “prey brought home” accurately reflects prey taken. Two studies using kittycams (USA & RSA) have reported that cats bring back only a fraction of what they kill, and more importantly, they are more likely to bring back some taxa than others. This needs to be acknowledged.
The English needs to be tidied up by an English speaker.
Line 45. Cats are actually one of the top 7 invasive species (see Bellard et al. Proc. R. Soc. B 2016) that globally threaten the highest numbers of mammals, birds, reptiles and amphibians.
Line 45. Invasive species are not native so no need to describe as non-native.
Line 68. In fact we don’t know why cats bring their prey home, but it is not necessarily as a gift. What we do know from the studies conducted in the USA and Cape Town is that less than a third of prey caught are brought home. This needs to be acknowledged.
Lines 71-72. The kittycam studies have shown that cats are more likely to bring back home some prey types over others, therefore, one can’t assume that the proportions of prey types brought home reflect the proportions of prey types eaten, as is asserted in this sentence. It is possible that certain types of prey are more likely to be consumed on site at certain times of the year.
Lines 84-86. This prediction isn’t new or an extension of what we already know about cat predation rates. The Thomas et al. paper cited [27] used cat owner records of prey brought home to show that predation was higher in spring and summer; therefore, it is not clear what new insights are provided by this study predicting exactly the same trend. Unless the authors have some good reason to believe it might be different in France.
Lines 90-92. Surely it would be the relative rainfall within a region that would be the potentially important predictor of prey activity, not the rainfall of different regions? If the latter, would it mean mice were perpetually more active in regions with higher rainfall?
Lines 94-97. This prediction seems overly simplistic. And what is meant by “increase”? Will the species richness of these groups increase in number in the prey brought home, or the abundance increase. Endothermic species can be well adapted to cooler climates, so it is not clear to me why fewer should be caught just because the climate is cooler.
Lines 99-100. What does the Human Footprint Index measure? The mechanism behind this prediction should be explained. Is it because areas of HFI have less green space and depleted animal communities?
Line 106.How were people recruited into the study?
How did you know how reliable cat owners were at reporting prey caught by their cats?
How long did cat owners report prey caught by their cats? Some stats on mean lengths of times owners reported over would be useful.
Line 155. Why was month a continuous variable? I would have thought it was a categorical variable.
Results
Please report the average number of prey brought home by cats over a standard length of time.
Line 198. Identifying rather than characterizing.
Table 1. The group of animals each model was investigating should be made much clearer in this table. At the moment it is a subscript of “GAMM”.
Figure 2. Labels on axes need to be larger and clearer.
Line 288. It is claimed here that cat owners intensively monitored prey brought home by their cats during eight years, but there is no measure of just how intensive this monitoring was – for example, how many prey were recorded for each cat.
Lines 313-315. I don’t think the results of these studies can be discounted due to the different sample sizes. The monitoring of the cats in the NZ studies was more comprehensive and intensive than the monitoring in this online study, where you don’t really know whether respondents were reporting all prey brought home.
Lines 321-322: these two observations aren’t really comparable. I.e. it is possible that 28.7% of a small number of bats admitted to rehabilitation centres were attacked by cats, and this doesn’t imply under-appreciation of predation of bats by cats.
Line 355. It seems unlikely that birds brought home in spring would be juveniles – these are more likely to be caught in summer or autumn. What else is going on in spring that might make birds more vulnerable to predation?
Section 4.3. Don’t these trends just reflect that cats take prey in proportion to their availability?
Section 4.4. Perhaps consider the impact of human-built infrastructure and human-provided food subsidies on resources and habitats available for these species.
Line 435. Explain why shrews are inedible prey.
Comments on the Quality of English Language
This needs to be proof-read by an English speaker; required changes are many but mostly just better choice of words.
Reviewer 2 Report
Comments and Suggestions for Authors
Dear Authors, you have made a huge work for this paper and you should be awarded for that, although some minor points are missing.
In the introduction you could mention that several birds protection associations recommend to keep all cats indoor for preventing their predatory behaviour impact on wild life. Those recommendations are very deleterious for the cats welfare.
Your paper provides a much more nuanced analysis.
The conclusion might give the readers some elements for a potential position statement.
Reviewer 3 Report
Comments and Suggestions for Authors
Dear Authors
Interesting article that needs some corrections, additions:
Please do not repeat sentences in the Simple Summary and Abstract sections-see lines 19-25 and 33-39. In the Introduction section, please provide the dating and the centers where the domestication of the cat occurred.
Is it known what breed the cats included in the experiment were? How many were typical rooftop cats vs. purebred cats? Figures 2 through 6 are unreadable as currently presented, please improve their quality/size. The study group is sufficient for reliable statistical calculations. The Discussion section is interestingly led. The Conclusion section is more of a Results section, so please be sure to improve this part of the paper.
Regards
Comments on the Quality of English LanguageMinor editing of English language required.
Round 2
Reviewer 1 Report
Comments and Suggestions for Authors
Review of: spatio-temporal and individual patterns of domestic cat hunting behaviour in France.
This revision is much improved on the initial version. I would have appreciated a file with replies to my comments, explaining why the authors opted to take them aboard or not.
I still question the logic behind prediction 2: there are regions of the world with very high rainfall and high levels of bird activity. If areas are wetter then the animals that live there are adapted to that climate. Rainfall is more useful as a predictor looking at its effect across different climatic conditions in the same region.
The English still needs to be improved – there are many small errors.
Line 129. Could the authors explain what “rooftop bred” means.
Line 313. I don’t think the authors can claim that cat owners closely monitored prey brought home by their cats over 8 years, because there are no data to support just how closely owners were monitoring their cats, and the statistics reported in the first paragraph of the results suggests owners were in fact not closely monitoring their cats over the full eight years, as frequency of prey brought home was very low.
Comments on the Quality of English LanguageEnglish need to be improved, but it shouldn't take an English speaker long to do this.
Author Response
Dear Reviewer 1,
We wish to thank you for the time spent on our manuscript.
We have made the minor changes on the revised version of the manuscript and we have carefully responded to each of the comments below (in blue).
Reviewer: 1
Comments to the Author
This revision is much improved on the initial version. I would have appreciated a file with replies to my comments, explaining why the authors opted to take them aboard or not.
- Thank you again for all your comments, which have enabled us to improve the manuscript considerably. However, we are sorry to read that the 5 pages of responses to your first review were not sufficient (page 1 to 5). We felt that all your comments had been answered and argued point by point.
I still question the logic behind prediction 2: there are regions of the world with very high rainfall and high levels of bird activity. If areas are wetter then the animals that live there are adapted to that climate. Rainfall is more useful as a predictor looking at its effect across different climatic conditions in the same region.
- We agree, so we have specified the geographical region associated with our hypothesis, see lines 91-92: “Weather conditions strongly influence small mammals [29–31], birds [32,33] and lacertids [34] activity in Europe […].”.
The English still needs to be improved – there are many small errors.
- We had the entire revised manuscript proofread by a native American researcher (Joseph LANGRIDGE), which largely explained the long delay in resubmitting our manuscript. We are not in a position to send the manuscript to him for re-reading in such a short time, and we doubt that he will be able to correct all these small errors if he did not catch them in first place.
Line 129. Could the authors explain what “rooftop bred” means.
- Now lines 128-130: “Most of the monitored cats in this study were described by their owners as rooftop bred (93%, i.e., generic domestic shorthair or longhair cats), thus we didn’t include this variable in our analysis.
Line 313. I don’t think the authors can claim that cat owners closely monitored prey brought home by their cats over 8 years, because there are no data to support just how closely owners were monitoring their cats, and the statistics reported in the first paragraph of the results suggests owners were in fact not closely monitoring their cats over the full eight years, as frequency of prey brought home was very low.
- We would like to bring your attention to the distribution of records of prey taken home: 3073 cat owners reported just one prey item (61%), while 2610 owners reported at least 2 preys (39%), including 812 owners with 10 to more than 50 preys (16%). This distribution explains why the average number of reports is low and the standard deviation higher than the average. However, individual cats were followed for an average of 3.1 months ± 5.0 months, which is not an insignificant period. But we now use regularly rather than closely, see line 313.
